

# Hydrologic implications of aerosol deposition on snow in High Mountain Asia rivers

Naoki Mizukami[1], Samar Minallah[1], Cenlin He[1], Chayan Roychoudhury[2], William Y.Y. Cheng[1], Rajesh Kumar[1]

[1]NSF National Center for Atmospheric Research, Boulder, CO, USA
[2]Department of Hydrology and Atmospheric Sciences, University of Arizona, Tucson, AZ, USA

*Correspondence to*: Naoki Mizukami (mizukami@ucar.edu)

**Abstract.** The deposition of light-absorbing particles (LAPs) on snowpack is known to accelerate snowmelt. However, the resulting hydrological impacts, particularly on streamflow, remain under-explored. This study assesses the hydrologic

consequences of LAP deposition on snow in High Mountain Asia (HMA) based on model simulations with and without aerosol deposition ("clean snow" scenario) during 2004-2018. We use the Community Land Model with a detailed aerosol-snowpack radiation module and the mizuRoute river model, driven by a 12-km meteorology-aerosol reanalysis dataset— the Model for Atmospheric Transport and Chemistry (MATCHA)—generated by the Weather Research and Forecast model coupled with Chemistry that assimilates satellite aerosol optical depth every three hours. The results show that LAPs advance seasonal snow

cover disappearance by two weeks to over a month compared to the clean snow scenario. This shift alters both runoff and evapotranspiration (ET). LAP-deposited snow produces more runoff and ET until it is depleted; afterward, runoff declines due to earlier loss of snowmelt, while elevated ET persists even after LAP-deposited snow disappears as a darker, snow-free surface enhances evaporation from soil. Consequently, the annual runoff is slightly reduced under the LAP-deposited snow condition. Streamflow increases from late winter until snow melts completely but decreases due to earlier snow disappearance in LAP-

deposited snow. This pattern is most evident in the headwaters, with the impact diminishing downstream. The semi-arid basins in the western HMA (e.g., Amu-Darya and Indus) show greater sensitivity to LAP deposition than the monsoon-dominated eastern HMA (e.g., Brahmaputra and Ganges). In western HMA regions, where larger perennial snow exists, LAP-enhanced snowmelt persists into summer and fall, leading to greater streamflow during these seasons compared to the clean snow scenario. This study provides important implications for the synergistic control of air pollution and water resource

management.

## 1 Introduction

Many of Central–South Asia's major rivers originate in High Mountain Asia (HMA), the Tibetan Plateau, and its surrounding mountain ranges, and support critical water resources for one of the most populated regions in the world (Barnett et al., 2005; Immerzeel et al., 2010). The hydrology of these river basins is complex, with processes varying greatly across regions and





seasons. While snow accumulation and glaciers are prevalent at high elevations, summer-to-fall monsoonal precipitation brings
      substantial rainfall to the central and eastern Himalayas, even at high altitudes. In contrast, hydrology becomes snow-glacier
      melt driven in the western HMA. Consequently, the relative contributions of rain, seasonal snowmelt, and glacial melt to the
      total runoff differ markedly across the region (Armstrong et al., 2019; Bookhagen and Burbank, 2010; Chandel and Ghosh,
      2021; Immerzeel et al., 2009). In the far western part of the region, such as the Indus River basin, the snowmelt contribution

to total runoff is highest in the region (>50% of total runoff), compared to approximately 25% in the Tibetan Plateau and less
      than 20% in the central Himalayas (Armstrong et al., 2019; Bookhagen and Burbank, 2010).

      Snow and glacier processes at high elevations are complicated because of topographic complexity that influences energy fluxes
      and, therefore, snowmelt across local to regional scales. Additionally, recent studies have increasingly reported the deposition
      of short-lived aerosols known as light-absorbing particles (LAPs), such as black carbon (BC), dust, and brown carbon (BrC),

in snowpacks and glaciers of the Himalayas through observations (Gertler et al., 2016; Gul et al., 2022; Qian et al., 2015; Yan
      et al., 2023). Once trapped in a snowpack, LAPs reduce albedo, absorb more radiative energy, and accelerate snow and ice
      melting (He, 2022; Skiles et al., 2018). Snow albedo feedback is also a major contributor to warming in the HMA (Guo et al.,
      2018; Ma et al., 2019; Nair et al., 2024) and is further amplified by LAPs through interactions between LAPs and surface snow
      and near-surface meteorology (Flanner et al., 2009; Qian et al., 2009, 2011; Xu et al., 2016). Spatial and temporal variabilities

in LAP deposition across the Himalayas have also been reported (Li et al., 2021; Sarangi et al., 2020). Dust tends to dominate
      deposition at high elevations in the western part of the region, whereas BC is more prevalent at lower elevations.

      Although the physical mechanisms of LAP-induced snowmelt acceleration are well understood through both observational
      and model studies (Flanner et al., 2007; Qian et al., 2011; Skiles et al., 2018), the integrated impacts of heterogeneous
      hydrometeorological conditions and LAP deposition on streamflow at large river basin scales remain underexplored. Most

previous studies have focused primarily on the effects of LAP deposition on snow cover alone (Roychoudhury et al., 2022,
      2025a; Xu et al., 2016) and on local scale snowmelt and runoff (Rahimi et al., 2019; Sarangi et al., 2020; Usha et al., 2022).
      The main objective of this study is to examine the effect of LAP-induced changes in snowpack processes on subsequent
      hydrological processes, including snowmelt runoff, evaporation, runoff, and ultimately, streamflow in large-scale river basins.

      We employ a land surface model including a detailed aerosol-snow albedo scheme and river routing model (Section 2.1),

forced by newly developed meteorology-aerosol reanalysis data (Section 2.2). To isolate hydrologic impact of aerosol
      deposition on snow, we perform two simulations: one with LAP deposition representing more realistic snow conditions
      presently, and the other without LAP deposition, namely "clean snow" scenario. By comparing these two simulations, we aim
      to answer the following scientific question: To what degree do snowmelt patterns affected by LAPs impact hydrologic fluxes
      (snowmelt, runoff, and evapotranspiration (ET)) and streamflow across the HMA river basins, and how do these flux changes

vary seasonally? Including streamflow for the analysis provides insight into how air pollution reduction in the areas could





change the seasonal streamflow pattern across the regions. This is important for water resource management not only in the mountains where snow cover exist, but also larger areas downstream.

## 2 Methods

### 2.1 Model descriptions

We use the Community Land Model version 5.1 (CLM5; Lawrence et al., 2019) for 12-km offline land surface simulations and the mizuRoute river model (Mizukami et al., 2016) to simulate streamflow. Both models are applied over the majority of Asia on a Lambert conformal grid bounded by 58° – 140° E and 4° – 40° N (Roychoudhury et al., 2025b). Our analysis focuses on four major HMA river basins (Fig. 1a) with distinct hydroclimate regimes: two summer-fall monsoon-affected basins (Brahmaputra and Ganges) and two semi-dry basins (Indus and Amu Darya).

CLM5 represents comprehensive land surface and subsurface hydrological processes that simulate vertical moisture storage and movement from vegetation to bedrock in each grid. At each grid cell, heterogeneous land cover types are represented by five land units (vegetation, lake, urban, crop, and glacier). Within the vegetation unit, 16 natural vegetation types, termed Plant-Functional Types (PFT), share a single soil column, but influence plant hydrology differently. Although CLM5 does not simulate glacier dynamics, the ice melt is computed for the prescribed glacier land unit. Full descriptions of CLM5 and its

enhancement over its predecessors are provided by Lawrence et al., (2019). The key features of CLM5 relevant to the snow processes in HMA include a multi-layer (up to 12 layers) snow model with maximum accumulation of 10-m Snow Water Equivalent (SWE), which enables better representation of perennial snow cover that exists in HMA, and the grain size of freshly fallen snow in CLM5 as a function of air temperature, which captures more accurate snow albedo in extremely cold environments. Other notable improvements in CLM5 are new parameterizations related to soil evaporation (Swenson and

Lawrence, 2014) and grid-specific soil depth down to the bedrock with explicit saturated and unsaturated zones in the subsurface, and an adaptive time-stepping numerical solution to Richard's equation to enable finer time stepping to avoid unstable or unrealistic (e.g., negative) soil moisture solutions.

CLM5 uses the Snow, Ice, and Aerosol Radiative (SNICAR) model (Flanner et al., 2007, 2009, 2021; He et al., 2024) to compute the albedo and radiative fluxes of the vertically resolved snowpack layers containing LAPs. SNICAR accounts for

snow-aging processes and meltwater transport that scavenges LAPs from the top layers to the bottom layers and redistributes LAPs through the snow layers (Flanner et al., 2007). Recent enhancements to SNICAR includes internal mixing of LAPs with snow grain (i.e., particles can reside inside ice crystals depending on growth mechanisms of precipitable hydrometeors (Flanner et al., 2012; He et al., 2019), non-spherical snow grain shape (He et al., 2017), a more accurate radiative transfer solver (Dang et al., 2019), and an improved optical property database (Flanner et al., 2021). Evaluation of the enhanced SNICAR in CLM5



leads to reduced biases in snow surface albedo, snow cover area, and snow water equivalent (SWE) in many parts of the globe, including the northwestern Tibetan Plateau (He et al., 2024).

For the mizuRoute river model (Mizukami et al., 2016), we set up a high-resolution, catchment-based river network from a subset of global MERIT-basins (Lin et al., 2019), comprising 51,182 catchments and associated river reaches in four major river basins: Brahmaputra, Ganges, Indus, and Amu Darya (Fig. 1b). MizuRoute is forced by total runoff (the sum of surface

runoff, subsurface runoff, and runoff from glaciers) from CLM5 at catchments defined in the MERIT-basin river network at a 1-hour time step using a diffusive wave routing scheme (Cortés-Salazar et al., 2023) to generate streamflow at each river reach. While river flow is primarily driven by gravity, diffusive wave routing accounts for the pressure gradient of the river water along the channel, which is an important physical mechanism for water movement in flat areas. An important parameter for diffusive wave routing is the Manning coefficient. We use a spatially constant value of 0.05, acknowledging the uncertainty

in its spatial pattern and magnitude.







**Figure 1: Four river basins in High Mountain Asia with 1000m elevation bands (top panel) and MERIT-basin river reaches used for the river model (bottom panel). Elevation grid in bottom panel is 12km used for WRF-chem forcing and CLM5 simulations. Triangle points are locations used for streamflow annual cycle plots in Fig. 7.**


## 2.2 Model forcing

Hourly surface meteorological variables and LAP deposition fluxes (see Supplementary Material S1 for the full list of the variables) are extracted from the 12-km long-term (2003-2018) aerosol-meteorology reanalysis dataset referred to as MATCHA ( Model for Atmospheric Transport and Chemistry in Asia; Kumar et al., 2024) to drive offline CLM5 simulations.

The MATCHA product uses the WRF-Chem model (version 3.9.1) coupled with CLM4.0-SNICAR. This WRF-Chem-CLM-SNICAR modeling system assimilates satellite observations of aerosol optical depth from the Moderate Resolution Imaging Spectroradiometer (MODIS) and carbon monoxide profiles from the Measurement of Pollution in the Troposphere (MOPITT) every three hours to constrain the representation of aerosols and chemistry and then simulates the interactions between atmospheric composition (trace gases and LAPs), radiation, clouds, snow, and land surface processes. The meteorological,

chemical, and land surface parameterizations in WRF-Chem are based on past efforts to simulate the key meteorological and chemical characteristics of South Asia (Kumar et al., 2015, 2013).

Detailed evaluations of surface meteorology and aerosols were performed by Roychoudhury et al., (2025); therefore, the results of the key variables are briefly summarized here. MATCHA captures spatial variations for 2-m temperature, precipitation and snow cover area, indicated by strong spatial correlation ($R$) compared to observational dataset for all four seasons ($R$>0.9, 0.7,

and 0.7 for temperature, precipitation and snow cover area, respectively). However, the bias patterns are unique to each meteorological variable. MATCHA exhibits cold bias at high elevation (5-10 C-degree) compared to ground-based stations from the NOAA's Integrated Surface Database (ISD), with the largest bias over the western Himalaya, Karakoram, and Pamir for all seasons, and weak warmer bias (<5 C-degree) over lower elevations, including the Indian subcontinent and central Asia, except during the summer months. Comparison of daily precipitation with NASA's Integrated Multi-satellitE Retrievals for

GPM (IMERG) shows overestimation (< 5 mm/day) during winter through spring in high elevation areas, including Himalaya and Karakoram. However, MATCHA captures localized features along mountain foothills, which are smoothed over in the IMERG. The uncertainty of the estimated spatial pattern and magnitude of surface aerosol transport and deposition is propagated to the snow albedo estimate in the land model (Yasunari et al., 2013), therefore it is important to understand aerosol forcing uncertainty. For surface Black Carbon, despite the limited number of ground measurements (19, 13, and 24 sites for

daily, monthly, and seasonal mean, respectively), MATCHA exhibits a seasonal pattern consistent with the observations over the domain, and strong spatial correlation with observations across all seasons (0.6-0.7), with the highest correlation observed in spring and winter (~ 0.75), while the lowest is observed in summer (~ 0.6). However, high-elevation regions, including the



Himalayas, consistently show underestimation by the model across all seasons. Further discussion of LAP radiative forcing is provided in Supplementary Material S2.

## 2.3 Model experiments

Two sets of offline CLM5 and mizuRoute simulations are conducted. The first, referred to as CLM-LAP, is driven by hourly meteorological and LAP deposition fluxes, including black carbon, dust, and brown carbon particles, provided by MATCHA. The second set, CLM-clean, is driven only by meteorological variables, and all LAP deposition fluxes are set to zero. Both experiments are run from 2003 through 2007 five times for the soil initializations and then run with the initialized soil states for a 16-year period from 2003 to 2018. The analysis uses simulations from 2004 to 2018.

A similar experiment was carried out by Usha et al., (2020, 2022), using a 50-km Regional Climate model (RegCM) that includes CLM4.5 with SNICAR LAP processes and a snow module to investigate the impacts of LAP on snowmelt and surface runoff in the Himalayan regions. The RegCM used by Usha et al., (2022) is an atmosphere-land coupled model; therefore, they examined aerosol effects on atmospheric radiation as well as albedo due to its impurities separately as well as combined effects on snow. The key novelty of our study compared to Usha et al., (2022) includes the following:1) use of the updated SNICAR, as detailed in Section 2.1; 2) higher grid spacing (12 km versus 50 km), resolving topography better; 3) 1-hour LAP deposition data generated using a state-of-the-art WRF-Chem modeling framework with data assimilation within MATCHA, as described in Section 2.2; 4) focus on key hydrologic fluxes (i.e., snowmelt, runoff, and ET); and (5) additional river routing simulations using mizuRoute to assess the impacts on the streamflow across the large river basins.

## 3 Results and Discussions

### 3.1 Impact of LAP deposition on snow cover

The impact of LAP deposition on snow cover is examined because the extent of snow cover plays an important role in hydrological processes. We categorize snow cover into four types for each year and for the multi-year (2004-2018) mean: no snow cover, ephemeral, seasonal, and perennial. If the snow water equivalent (SWE) in a given area never exceeds 5 mm for a given year, that area is marked as having no snow cover. Seasonal snow cover is defined as snow (with greater than 5 mm of SWE) on the ground that persists for 60 days or longer continuously per year; otherwise, areas are considered ephemeral snow cover. If the SWE is always above 5 mm throughout the year, these areas are marked as perennial snow cover. These definitions, although arbitrary, are equivalent to those used in a western U.S. snow cover study (Petersky and Harpold, 2018). For seasonal snowpack, we compute snow appearance day and disappearance day for each year and then 2004-2018 mean values are shown for CLM-LAP and CLM-clean runs in Fig. 2 (a, b, d, e). Fig. 2g and 2h compare the mean snow appearance and disappearance between the two cases (CLM-LAP and CLM-clean, respectively). LAP-deposited snow cover appears seven days later in 6% of snow cover areas and disappears seven or more days earlier in 23% of snow cover areas than in clean snow



cover areas. Given that the snowfall amount and the other meteorological conditions for both simulations are identical, the LAP-deposited snowpack melts quicker in the earlier snow accumulation season than the clean snowpack before seasonal
snow cover is established. As shown in Figs 2c and 2f, CLM-LAP has smaller areas defined as perennial snow cover than CLM-clean, approximately 40% less perennial snow than clean snow conditions. The hydrological processes over perennial and seasonal snow cover are quite different, as discussed in the following section.

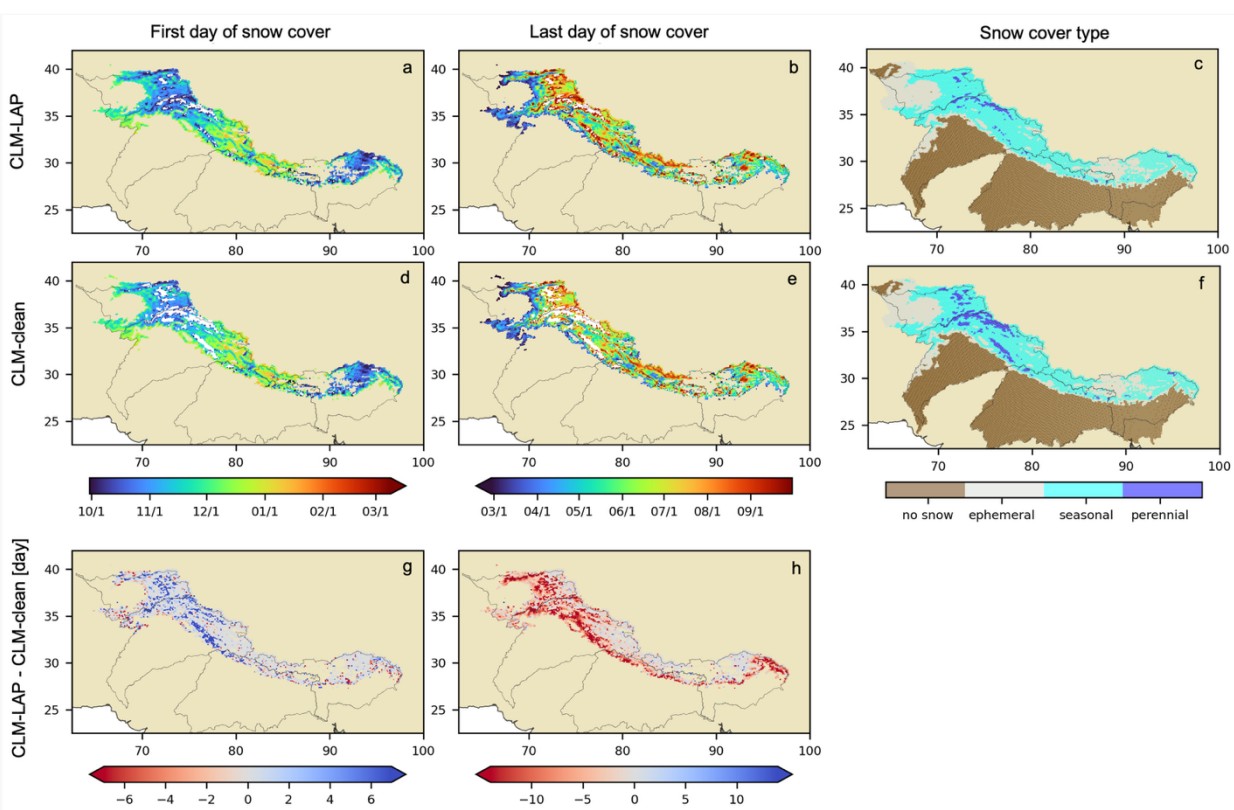

**Figure 2: 2004-2018 mean 1st day of snow cover for CLM-LAP (a) and CLM-clean (d) and the last day of snow cover for CLM-LAP
(b) and CLM-clean (e) and their difference between CLM-aerosol and CLM-clean (panels g and h). White area is perennial snow cover area. Positive values of day difference indicate first and last days of snow occurs later times for CLM-LAP. Right column (panels c and f): snow cover type-no snow cover (<5 mm always), ephemeral, seasonal (>5 mm for 60 or longer days continuously) and perennial snow covers (>5 mm always).**

### 3.2 Impact of LAP deposition on hydrologic fluxes

Figure 3 shows the absolute differences in 2004-2018 mean annual and seasonal water fluxes, including rain plus snowmelt, total runoff (subsurface runoff plus surface runoff) and ET, between the CLM-LAP and CLM-clean. For this analysis, we use rain plus snowmelt as the total water flux into the terrestrial hydrological system instead of precipitation because perennial snowpack exists at high elevations of HMA, where snowfall can be stored as snowpack for years instead of contributing to runoff in each year. Because the rainfall amounts are identical for both CLM-LAP and CLM-clean, the difference in rainfall





and snowmelt is due to changes in snowmelt runoff; therefore, the term of snowmelt is used hereafter when mentioning the difference in rain plus snowmelt between the CLM-LAP and CLM-clean. Also, Fig. 3 focus two snowmelt seasons, spring (MAM) and summer (JJA), but Supplemental material S3 shows seasonal water fluxes for CLM-LAP and relative and absolute difference between the two simulations for all the seasons including winter (DFJ) and fall (SON).

Figure 3 a shows that the annual snowmelt amount is very similar for both CLM-LAP and CLM-clean over the seasonal and
ephemeral snow cover areas (indicated by large grey areas), despite of the seasonal differences in snowmelt exist (Fig. 3 d and g), indicating consistent annual water availability for both LAP deposited snow and clean snow. This is not the case for the areas of perennial snow cover in CLM-LAP where persistent and greater snowmelt occurs than clean snowpack. There are small (by <10 mm/month) increase in LAP snowmelt in the ephemeral snow cover area during winter (DJF; see Fig. S3-2). This increase is balanced by the decrease in snowmelt during spring (MAM) for LAP-deposited snow compared to clean snow
(light red shaded area in Fig. 3 d).

Figure 3 b and c indicate overall the annual partitioning of rain plus snowmelt into runoff and ET slightly changes over seasonal snow cover areas; decreases in runoff and increases in ET for LAP-deposited snow cover areas by a similar magnitude (40 mm/yr or less) compared to the clean snow scenario. In the perennial snow cover areas (in Karakoram and Pamir), however, both annual runoff and ET increase in CLM-LAP compared to CLM-clean. This is because LAP deposited snowpack produces
greater snowmelt through the year from the almost unlimited amount of snow storage in perennial snow (Fig. 3 a).

The seasonal runoff and ET differences shown in Fig. 3 e-f and h-i indicate that runoff increases over the LAP-deposited snow cover in spring but decreases in summer to a greater degree than the magnitude of increase in Spring, resulting in a decrease in annual runoff. However, the ET increases during both spring and summer (except at lower elevations). Qian et al., (2011) also shows BC-deposited snowpack increases total runoff during spring and switches to reduction during summer in the
Tibetan Plateau. This difference in seasonal runoff and ET changes, as seen in the modeling experiment on dust-on-snow impacts on hydrological fluxes in the Colorado River (Painter et al., 2010), can be explained by the difference in snow-covered days between LAP-deposited snow cover and clean snow cover.



## Water flux differences (LAP snow – clean snow)



**Figure 3: Top panels: Difference in 2004-2018 mean annual snowmelt runoff (left column), total runoff (middle column) and**
**evapotranspiration (right column) between CLM-LAP and CLM-clean [mm/yr]. Middle and bottom panels: Difference in spring (MAM) and summer (JJA) snowmelt runoff, total runoff and evapotranspiration between CLM-LAP and CLM-clean [mm/month].**

Figure 4 illustrates the differences in snow disappearance days and how this affects soil moisture, ET, and runoff in a selected grid box where both CLM-LAP and CLM-clean produce seasonal snow cover. When both LAP-deposited snow and clean snow exist (orange shaded area), LAP-deposited snow cover produces more snowmelt owing to greater radiative forcing,
causing more soil moisture recharge, ET, and runoff than clean snow. When LAP-deposited snow cover disappears while clean snow cover still exists (blue shaded area), CLM-LAP continues to produce more ET than CLM-clean owing to the darker surface exposed than the snow-covered area in CLM-clean, resulting in more energy absorption than the snow cover area. The ET in the CLM-LAP is produced by the expenditure of soil moisture. Because no snowmelt supplies additional water flux in CLM-LAP after the snowpack disappears, runoff is also reduced, while CLM-clean still produces runoff from snowmelt. When
both LAP-deposited and clean snow disappear, both produce similar ET and runoff.







**Figure 4: Daily time series of SWE and soil moisture, ET and runoff at a grid box of latitude 38.59o N and longitude 71.20o E located in upper Indus River from 2014-04-01 to 2014-09-30. Orange shade indicates snowpack exists on ground for both CLM-LAP and CLM-clean, while blue shade indicates only CLM-clean has snowpack.**




### 3.3 Impact of LAP deposition on streamflow

The impacts of LAP deposition on the timing and magnitude of streamflow are assessed at each river reach, including downstream reaches of the river basins, since streamflow is the cumulative runoff from all upstream areas. Metrics that can be used to quantify the timing of snowmelt-driven streamflow are 1) the centroid (i.e., center of mass) of annual hydrographs, 2)

annual peak dates, and 3) snowmelt pulse onset (Stewart et al., 2005). The annual hydrograph centroid provides a less noisy, volume-integrated perspective of the timing of the snowmelt runoff pulse and the overall distribution of streamflow for each year compared to the other two metrics (Stewart et al., 2005). In fact, the impacts on the annual peak flow timing are less than a few days at maximum (not shown).

Figure 5 shows the 2004-2018 mean annual peak flow and centroid of the 2004-2018 mean annual hydrograph and the

differences between CLM-LAP and CLM-clean per river reach in the three basins. The simulated 2004-2018 mean peak flows downstream of all three rivers range from 30,000 $m^3/s$ (Indus River), 60,000 $m^3/s$ (Ganges), to 80,000 $m^3/s$ (Brahmaputra). A brief validation of simulated streamflow at the outlets of the Brahmaputra and Ganges Rivers is provided in Supplementary Material S4; however, a full evaluation of streamflow is beyond the scope of this study. The impact of the LAP-deposited snowpack on the annual peak flow is small, and the relative difference is less than 6% compared to the clean snow case for the

eastern part of the HMA (i.e., Ganges and Brahmaputra). For Amu Darya and Indus, larger differences are observed in the upstream areas, but the sign of the difference is mixed. Peak flow increases where the drainage areas have a perennial snowpack, and a reduction of peak flow is seen elsewhere. The center of mass in the annual hydrograph in the LAP-deposited snow case shifts earlier, indicating that a larger volume of streamflow occurs earlier in years than clean snow. This is because the enhanced snowmelt in late winter through early spring contributes to the river flow earlier in the season. However, the

timing shift is, at most, within a week.





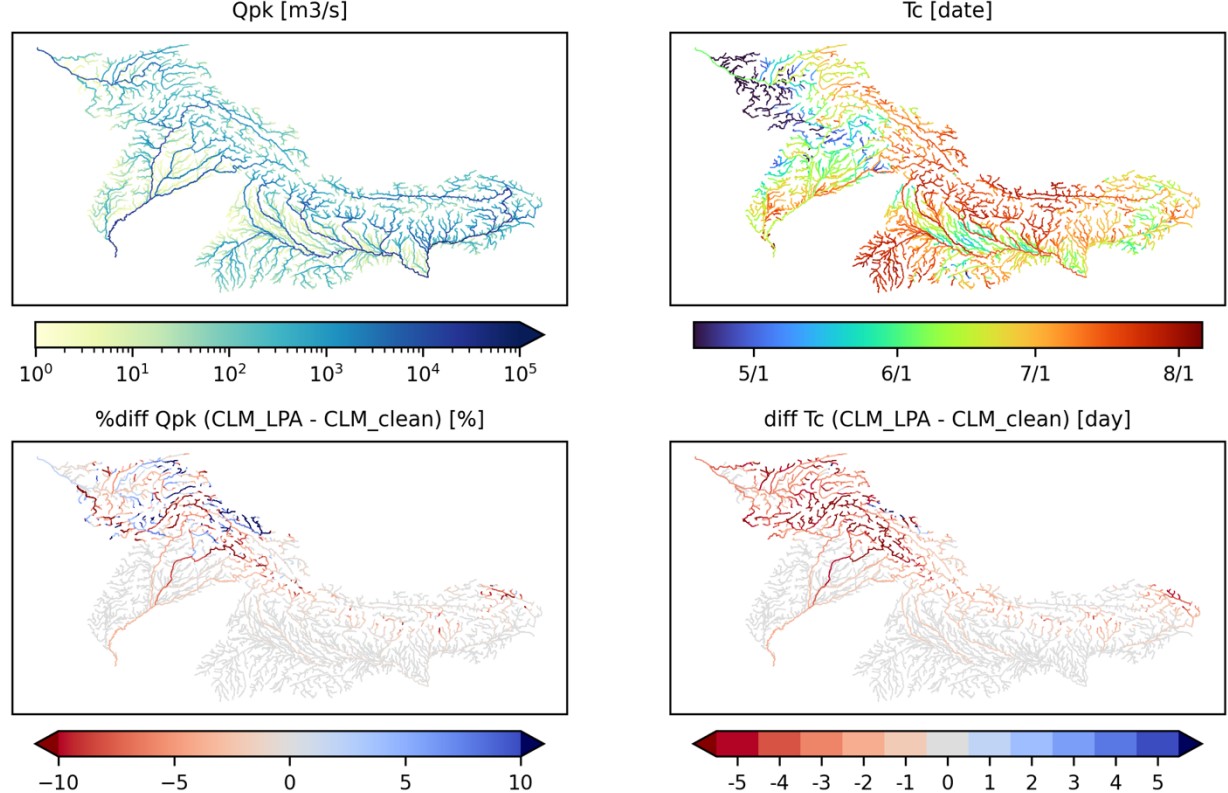

**Figure 5: 2004-2018 mean peak discharge and seasonal hydrograph centroid and their changes (bottom). Negative values indicate earlier times.**

Figure 6 shows the 2004-2018 mean annual cycle of the simulated streamflow at the outlet of each basin and one upstream
location of that basin (See Fig.1) and the absolute differences between CLM-LAP and CLM-clean (right column). LAP
deposition on snowpack results in an increase in streamflow during late winter through spring, which then decreases later in
the season, although the magnitude of the difference is small, as indicated by the peak flow difference in Fig. 5. The timing of
this shift from the streamflow increase to decrease does not necessarily correspond to the peak time, but rather to the difference
in snow disappearance days. Figure 6 indicates that the impacts of LAP in the snowpack on streamflow are relatively small
downstream of Ganges and Brahmaputra. This is expected because a large portion of the river water sources are from rainfall
at low elevations when the entire Ganges and Brahmaputra are considered. Larger differences were noted for Indus and Amu
Darya.







**Figure 6: Left panels: 2004-2018 mean annual cycle of streamflow at outlet of the river basin (solid line) and example of upstream**
**location (dash). Right panels: difference in mean annual cycle of streamflow between CLM-LAP and CLM-clean (orange minus red). Orange is CLM-LAP and Blue is CLM-clean on left panels.**

Figure 7 shows the largest increase and reduction of annual streamflow cycle in the LAP deposited snow case compared to the clean snow case at individual river reaches and their timings. The largest increase in streamflow occurs earlier or before the
peak time, whereas the largest streamflow reduction occurs around or after the peak time. Although the magnitudes of the streamflow increase and reduction are greater downstream of the basin than upstream as the flow difference at each reach





accumulates from upstream to downstream, the relative differences are greater upstream or in mountainous headwater catchments. This indicates that the impacts on seasonal water supply become greater in the upstream area of the basins because of significant changes in streamflow relative to the case of the clean snowpack.


**Figure 7: Maximum flow increases (LAP minus clean) due to LAP deposition in snow (left panels) and flow reduction (right panels) based on 2004-2018 mean annual streamflow cycles. Top panels are in m3/s and middle panels are %change relative to clean snow scenario. Bottom panels show timing of flow increase and reduction. River reaches of stream order of 3 or greater are shown.**





## 4 Conclusions

The impacts of LAP in the snowpack on hydrological fluxes, including streamflow over the HMA originated basins, are assessed based on 12-km simulations using CLM5 coupled with the enhanced SNICAR snowpack radiation model and mizuRoute river model, driven by newly developed long-term (2003-2018) hourly MATCHA meteorology-aerosol reanalysis 275 data.

Spatial-temporal changes in runoff and ET due to LAP deposition on snowpack are explained by differences in snow cover types (seasonal or perennial) and snow cover disappearance days. Therefore, it is important to characterize the extent and type of snow cover. LAP deposition on snow affects the extent of the perennial snow cover. LAP deposition increases snowmelt runoff during the winter accumulation period through the ablation period by as much as 50% compared with that of clean 280 snow. This leads to greater runoff and ET during the same period. After snowmelt ceases earlier in LAP-deposited snow due to the earlier disappearance of snowpack, runoff drops, but larger ET is sustained from soil moisture. Larger runoff and ET persist in summer and fall over the perennial snow-covered area.

The magnitude of the streamflow change due to LAP deposition on snow becomes larger downstream in the basins as rivers accumulate runoff, but the significance of the impacts is greater upstream in basins in proximity to snow-covered areas. The 285 peak flow is not necessarily the most affected by the altered snowmelt pattern due to LAP deposition, as the shift in the timing of the annual peak discharge is within a few days. The overall increased streamflow occurs earlier upstream of the Indus River and Amu Darya to a greater degree than Ganges and Brahmaputra.

The tracer analysis of the LAPs in the HMA shows that the burden of BC peaks in winter, with China and India as the dominant sources of the anthropogenic BC (Roychoudhury et al., 2025). The regions surrounding the HMA, particularly South Asia, 290 experience some of the most severe air pollution in the world, underscoring urgency of effective air pollution management (UNEP, 2019). We hope this study offers useful insight into how reducing BC through air pollution control measures could influence water resources, not only locally snowpack-dominated areas, but also across larger downstream regions with dense population. We acknowledge the uncertainties in the model simulations arising from the forcing (Roychoudhury et al., 2025) and CLM5 parameterizations, and the improved model accuracy is critical for the practical water resource managements. 295 Recent research on the computationally expensive, complex land-model calibration (Cheng et al., 2023; Elkouk et al., 2024; Tang et al., 2025) shows promise in improving the model accuracy of the targeted variables; however, these efforts are left for future work.

Finally, one follow-up question is the relative contribution to the hydrologic shift between LAP deposition on snowpack and climate trends and variability. LAP radiative forcing can be as significant as greenhouse gas-induced warming for snowmelt 300 (Flanner et al., 2007; Qian et al., 2015; Ramachandran et al., 2023; Yasunari et al., 2013). Although wet, rainfall-dominant



basins are impacted to a lesser degree, this may change in the future owing to a shift in climate conditions that affect mountain snowmelt processes as well as LAP emission changes.

*Code and data availability.* CLM5 forcing data, MATCHA (Kumar et al., 2024), is available at
https://doi.org/10.5067/CG4OT8DJX2Z7. The CLM5 and mizuRoute outputs are available upon request. The Python code to reproduce all figures will be available at https://github.com/NCAR/HMA_hydrology.

*Author contributions.* NM contributed to the conceptualization, methodology, formal analysis, investigation, visualization, writing- original draft, reviewing, and editing. SM contributed to the conceptualization, visualization, validation, writing-
review & editing. CH contributed to the methodology related to modeling, validation, project administration, funding acquisition, writing-review & editing. CR contributed to the data curation, validation, writing-review & editing. WC contributed to the data curation, validation, writing-review & editing. KR contributed to the funding acquisition, project administration, validation, writing-review & editing.

*Competing Interests*. The authors declare that they have no competing interest.

### Acknowledgement

This work was funded by NASA HiMAT2 grant through the Grant 80NSSC20K1342 and was performed at the NSF National Center for Atmospheric Research (NCAR), which is a research center sponsored by the U.S. National Science Foundation (NSF) under Cooperative Agreement No. 1852977. CLM5 and mizuRoute simulations were conducted using the Derecho
system (doi:10.5065/qx9a-pg09) provided by NSF NCAR's Computational and Information Systems Laboratory and sponsored by the NSF. The authors thank Dr. Sean Swenson for fruitful discussions.

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
