# Peer review of "Hydrologic implications of aerosol deposition on snow in High Mountain Asia rivers"

_EGUsphere, 2025_

## Referee Comment (RC1)

**Hydrologic implications of aerosol deposition on snow in High Mountain Asia rivers**

This study evaluates the hydrologic impacts of LAP deposition on snowmelt and streamflow across High Mountain Asia. The work is timely, important, and technically strong, and it addresses a critical gap in our understanding of how aerosols influence mountain water resources. The modeling framework is thoughtfully designed, and the manuscript is generally well executed. My comments below aim to strengthen clarity, methodological transparency, and interpretation of the results.

**Section 1 Comments:**

**Lines 40-41**: A line about the origin of LAPS would make this line complete, as understanding it is essential for this paper.

**Line 58-60**: I would recommend simplifying the research question; maybe breaking it into two research questions would help readers.

**Section 2.1 Comments:**

The section could benefit from clearer structuring. As written, it tries to introduce the study region, CLM5 physics, SNICAR processes, and the routing model all in one continuous block, which makes it difficult for the reader to retain the key points. I would recommend separating this into two subsections: Study Site (use Figure 1) and Model Setup. Including a simple conceptual diagram would also help the reader's understanding.

**Line 73-74:** The statement that CLM5 "does not simulate glacier dynamics" but includes glacier melt requires clarification. Specifically, how is meltwater from the prescribed glacier land unit treated and passed to mizuRoute? Since glacier-fed basins dominate HMA hydrology, it is important to know whether glacier melt is handled differently from seasonal snowmelt and whether this distinction has implications for routing or streamflow interpretation.

**Line 99-100:** Please justify the use of a spatially constant Manning coefficient (0.05) in mizuRoute. The four basins span extremely diverse channel geometries, from steep headwaters to broad lowland floodplains, so a single uniform value may not represent true hydraulic variability. Even a brief sensitivity analysis or supporting reference would help address this potential source of uncertainty.

**Section 2.2 Comments:**

A major concern is the substantial cold bias (5-10°C) at high elevations. This is not a small discrepancy, it directly influences precipitation phase, snowpack accumulation, melt timing, and therefore the size of the LAP signal you are trying to quantify. The manuscript should explicitly

discuss how this cold bias propagates into CLM5 and whether it potentially amplifies or dampens LAP-induced changes in snowmelt and runoff.

**Section 2.3 Comments:**

Please confirm explicitly that the meteorology is identical between the LAP and clean-snow experiments. Since this is an offline land-only setup, any difference in meteorological forcing would undermine the interpretation that differences between the two experiments arise solely from LAP radiative effects on the snowpack and hydrology.

The manuscript should also provide more justification for why a 5-year spin-up is sufficient for these high-elevation, cold-region basins.

Finally, the novelty bullets are useful, but they could be more clearly connected to why these differences (e.g., updated SNICAR, 12 km grid spacing, hourly aerosol deposition) actually improve hydrologic representation relative to Usha et al. Complexity does not automatically lead to better results; rather, it is important to explain how these specific model enhancements translate to more realistic snowmelt, energy balance, or streamflow behavior in HMA.

**Section 3.1 Comments:**

159-160: How are you computing the snow appearance or disappearance date? This should be clarified somewhere in the methods section.

**Section 3.2 Comments:**

**Line 184 ("very similar"):** The phrase very similar is vague. Please quantify the similarity (e.g., difference < X mm/yr or <Y%) so readers understand the actual magnitude. In general, whenever you use terms like "similar" or "slightly," providing numbers will strengthen the interpretation.

The text should explain a bit more clearly how melt behavior differs between ephemeral and perennial snow regions. My understanding is that:

- In ephemeral snow areas, the annual melt magnitude and timing is essentially the same between CLM-LAP and CLM-clean.

- In perennial snow areas, LAPs cause melt timing shifts. Please emphasize this as it is an important topic given the changing weather patterns.

**Figure 3:**

- Add clear titles colorbar indicating the variable and units.

- Use panel labels (a, b, c, etc.) consistently in the caption to improve readability (Recommend doing this for Figure 4 as well).

- Consider adding contours or masks (or anything ) indicating high-elevation/perennial snow zones. This would help relate spatial patterns to snow regime differences discussed in the text.

**Section 3.3 Comments:**

**Line 222- 225:** The section describes the use of hydrograph centroid, peak flow date, and snowmelt pulse onset as timing metrics, but it is unclear whether all three were actually computed. The text suggests only centroid and peak flow were used. The manuscript states that the impacts on annual peak flow timing are "less than a few days," but no quantitative value is given. Providing the actual range would help to know hydrologic significance.

**Line 232- 233:** The validation of streamflow is limited to a brief note in the Supplement, yet streamflow is the final outcome of the modeling chain. Even a short validation summary in the main text (e.g., bias or NSE at outlets) would strengthen confidence in the results.

**Line 230- 232:** For peak discharge magnitudes, the numbers reported (30,000–80,000 m³/s) are extremely large compared to typical observed flows. Please confirm whether these values refer to modeled total reach flow, unit-scaled routing output, or a specific model convention.

**General comment for this section:** When describing the peak flow differences, terms like "small," "mixed," or "larger differences" would benefit from numerical context.

**Line 239-240:** The finding that the hydrograph centroid shifts earlier but "within a week" is reasonable, but it would be helpful to relate this shift to the corresponding shift in snow disappearance timing shown earlier. Even briefly referencing the SDD change (e.g., "consistent with the earlier SDD by X days…") would help connect the snow and streamflow responses.

**Figure 5** requires clearer panel labeling and explicit identification of units on colorbars. It would also help to state whether negative centroid shifts consistently correspond to upstream perennial snow areas, as the text suggests but does not explicitly demonstrate.

**In Figure 6,** the interpretation that LAP impacts downstream are small because rainfall dominates the hydrograph is reasonable. It may be worth briefly noting the percentage of flow contribution from snowmelt vs. rainfall for each basin (even approximate, as I understand that it is out of scope), as this would help contextualize basin-to-basin differences.

**General Comments on the Conclusion:**

The conclusion introduces new concepts and external results (e.g., BC source attribution, winter BC burden, regional air pollution severity) that were not discussed earlier in the Results or Discussion sections. These points, while important, should appear in the main discussion or introduction, not be introduced for the first time in the conclusion. The conclusion should synthesize findings, not introduce new topics thus largely needs to be rewritten.

- The statement about China and India as dominant anthropogenic BC sources (via tracer analysis) appears abruptly. If the authors want to include this, it should be tied to the hydrologic findings earlier in the manuscript; otherwise, it reads like a new result with no preceding context.

- The section states that LAPs can alter snowmelt by up to 50% and shift streamflow timing, but does not discuss whether these magnitudes fall within or exceed model uncertainty bounds. Without framing results relative to uncertainty, the practical significance may be overstated.

Overall, the conclusion reads more like a mini discussion. I suggest:

- Moving background or new information (BC sources, air pollution severity, LAP vs. climate forcing) to the discussion section.

- Keeping the conclusion focused on synthesizing the modeling results:

  - what was learned,

  - what the key hydrologic implications are, and

  - what the main uncertainties and limitations are.

- Finally, the conclusion rarely acknowledges that all findings are model-based (CLM5 + SNICAR + mizuRoute + MATCHA forcing), it may help to explicitly remind readers that the results represent modeled LAP impacts., not observationally-derived hydrologic changes. Clarifying this helps contextualize applicability and avoids overstating confidence in the hydrologic responses shown.